# Association of Standardized Parenteral Nutrition with Early Neonatal Growth of Moderately Preterm Infants: A Population-Based Cohort Study

**DOI:** 10.3390/nu16091292

**Published:** 2024-04-26

**Authors:** Laurence Fayol, Jilnar Yaacoub, Marine Baillat, Clotilde des Robert, Vanessa Pauly, Gina Dagau, Julie Berbis, Frédérique Arnaud, Audrey Baudesson, Véronique Brévaut-Malaty, Justine Derain-Court, Blandine Desse, Clara Fortier, Eric Mallet, Anaïs Ledoyen, Christelle Parache, Jean-Claude Picaud, Philippe Quetin, Bénédicte Richard, Anne-Marie Zoccarato, Anne-Marie Maillotte, Farid Boubred

**Affiliations:** 1Department of Neonatology, Hôpital La Conception, Hôpitaux Universitaires de Marseille, 13005 Marseille, Franceclotilde.desrobert@ap-hm.fr (C.d.R.); farid.boubred@ap-hm.fr (F.B.); 2Réseau Périnatal Méditerranée (PACA-Corse-Monaco), 13015 Marseille, France; maillotte.am@chu-nice.fr; 3Research Unit EA 3279, Department of Public Health, Aix-Marseille University, 13005 Marseille, France; vanessa.pauly@ap-hm.fr (V.P.); julie.berbis@ap-hm.fr (J.B.); 4Centre Hospitalier de Martigues, 13500 Martigues, France; 5Clinique L’Etoile, 13540 Puyricard, France; 6Centre Hospitalier du Pays d’Aix, 13100 Aix-en-Provence, France; 7Centre Hospitalier Universitaire Nord, 13015 Marseille, France; veronique.brevaut@ap-hm.fr; 8Hôpital Saint Joseph, 13008 Marseille, France; jderain@hopital-saint-joseph.fr; 9Centre Hospitalier, 06130 Grasse, France; b.desse@ch-grasse.fr; 10Centre Hospitalier, 06600 Antibes, France; 11Centre Hospitalier, 20600 Bastia, France; 12Centre Hospitalier, 20090 Ajaccio, France; anais.ledoyen@ch-ajaccio.fr; 13Centre Hospitalier, 13300 Salon, France; 14Centre Hospitalier, 06400 Cannes, France; 15Centre Hospitalier, 84000 Avignon, France; pquetin@ch-avignon.fr; 16Centre Hospitalier, 83056 Toulon, France; bdeceaurriz@var.fr; 17Centre Hospitalier, 05000 Gap, France; anne-marie.zoccarato@chicas-gap.fr; 18Centre Hospitalier Universitaire, 06000 Nice, France; 19Aix-Marseille Université, C2VN, INRAE, INSERM, 13005 Marseille, France

**Keywords:** premature infant, parenteral nutrition, growth, moderately preterm, energy, protein, standardized parenteral nutrition

## Abstract

In preterm infants, early nutrient intake during the first week of life often depends on parenteral nutrition. This study aimed to evaluate the influence of standardized parenteral nutrition using three-in-one double-chamber solutions (3-in-1 STD-PN) on early neonatal growth in a cohort of moderately preterm (MP) infants. This population-based, observational cohort study included preterm infants admitted to neonatal centers in the southeast regional perinatal network in France. During the study period, 315 MP infants with gestational ages between 32^0/7^ and 34^6/7^ weeks who required parenteral nutrition from birth until day-of-life 3 (DoL3) were included; 178 received 3-in-1 STD-PN solution (56.5%). Multivariate regression was used to assess the factors associated with the relative body-weight difference between days 1 and 7 (RBWD DoL1-7). Infants receiving 3-in-1 STD-PN lost 36% less body weight during the first week of life, with median RBWD DoL1-7 of −2.5% vs. −3.9% in infants receiving other PN solutions (*p* < 0.05). They also received higher parenteral energy and protein intakes during the overall first week, with 85% (*p* < 0.0001) and 27% (*p* < 0.0001) more energy and protein on DoL 3. After adjusting for confounding factors, RBWD DoL1-7 was significantly lower in the 3-in-1 STD-NP group than in their counterparts, with beta (standard deviation) = 2.08 (0.91), *p* = 0.02. The use of 3-in-1 STD-PN provided better energy and protein intake and limited early weight loss in MP infants.

## 1. Introduction

In the last few decades, much attention has been focused on optimizing the nutritional support of premature infants. Early nutrition is an important determinant of neonatal growth, neonatal morbidity/mortality, and psychomotor development in preterm infants [1,2]. Parenteral nutrition (PN) is an essential component of nutritional management in these critical patients during the early weeks of life, especially when enteral nutrition is contraindicated or insufficient [3]. Standardized solutions or individual preparations can be used for PN. Standardized PN is typically commercially formulated to meet the nutritional needs of most infants. These solutions can be two-in-one solutions (amino acids [AA] and dextrose), or multi-chamber three-in-one bag solutions providing AA, dextrose, and lipids [4]. When using two-in-one standardized PN, intravenous lipids can be administered separately to meet nutritional demands. Standardized parenteral solutions offer various advantages, including stability and quality control, limited ordering, compound errors, and cost benefits [4,5,6]. Conceptually, individual PN can be better adapted to the specific needs of each patient. Nevertheless, standardized PN increases the quality of delivered PN; reduces errors related to prescription, preparation and administration; and is currently available [5]. However, the optimal formulation is unknown, and different solutions may be necessary to meet nutritional demands over the neonatal period.

Compared with those born at full term, moderately preterm (MP) infants are at higher risk of mortality and morbidities [7]. Thus, MP infants are a neglected population at risk of postnatal growth failure, which can reach 30% incidence [8,9]. Postnatal growth failure commonly begins as early as the first week of life and results from energy and protein insufficiency [9]. Nutritional management guidelines for this vulnerable population, which often requires PN during the first few days of life, are incomplete and inconsistent [10,11,12,13]. In this population, PN is generally administered via a peripheral venous line for a short period of time. In this situation, parenteral solutions are commonly low in concentration and provide a low macronutrient intake. Studies have focused on populations of very and extremely preterm infants; however, only a few have investigated the influence of standardized PN on neonatal growth in MP infants. An early PN strategy may help optimize nutritional intake and prevent early neonatal weight loss after birth.

This study aimed to evaluate the influence of standardized PN using three-in-one double-chamber solutions on early neonatal growth in a cohort of MP infants who required PN from birth onwards.

## 2. Materials and Methods

### 2.1. Study Design and Population

This population-based, retrospective, observational cohort study was part of a regional healthcare quality-improvement project involving preterm infants admitted to neonatal units in the PACA-Corse-Monaco region in southeast France. Parents were informed of the study and could oppose the collection of data on their infants. The infant records were anonymized for analysis. This study was approved by the Commission Nationale de l’Informatique et des Libertés and Scientific Committee of the Regional Perinatal Health Network (number 1979318). We included all MP infants born between 1 January 2017 and 31 December 2017, with a gestational age (GA) from 32^0/7^ weeks to 34^6/7^ weeks and who received parenteral nutrition (PN) from birth to at least the third day of life (DoL3). Infants who died prior to discharge from the neonatal unit, those with major congenital and genetic abnormalities, and those with missing data on early nutrient intake and growth parameters were excluded from the analyses.

### 2.2. Principal Outcome: Early Neonatal Growth

In our study, early neonatal growth was defined as the relative body weight (BW) difference between birth (DoL1) and DoL7 (RBWD DoL1–7) and was calculated as follows: (DoL7 BW—birth BW)/birth BW (expressed as a percentage).

### 2.3. Nutrition Parameters

For this study, we compared two PN strategies based on the types of PN formulations used: (1) ready-to-use 3-in-1 standardized PN (3-in-1 STD-PN) and (2) commercially available pharmaceutical formulations with marketing authorization for neonates (Other-PN solutions). We conceptualized 3 different 3-in-1 STD-PN formulations to meet the nutritional needs of most infants according to postnatal growth phases, nutritional needs, and the type of venous line access (Appendix A). The 3-in-1 STD-PN solutions were formulated in cooperation with regional neonatologists to optimally fit the nutritional needs of preterm infants; they were prepared by a pharmaceutical manufacturer (Baxter Façonnage©, Montpellier, France). These included AA, lipids (SMOF lipids©), glucose, electrolytes, and micronutrients at different concentrations in two-chamber bags mixed at the bedside. Commercial pharmaceutical formulations used during the study period consisted of simple PN solutions (glucose with electrolytes or isolated AA solutions) or a 2-in-1 solution (AA + glucose and electrolytes); lipid emulsions, vitamins, and micronutrients could be administered separately.

Enteral nutrition and PN intakes were collected and calculated on days 1 (DoL1), 3 (DoL3), and 7 (DoL7). In this study, nutritional management was left to the discretion of the medical staff of each neonatal unit. We also collected information on the type of venous access (use of central line perfusion or peripheral access) and duration of PN.

### 2.4. Data Collection

Obstetric and neonatal data were collected from the infants’ medical records. The GA was calculated based on the date of the mother’s last menstrual period and/or early obstetric ultrasound findings. Small for gestational age (SGA) was defined based on gestational age and sex according to Fenton growth curves (https://peditools.org/fenton2013 accessed on 10 March 2018). Neonatal morbidity included severe respiratory distress syndrome (RDS), defined as the need for endotracheal surfactant therapy; necrotizing enterocolitis ≥ stage II according to Bell’s definition; early-onset infection (in the first three days) defined as at least one positive blood culture associated with clinical manifestations or antibiotic use for at least 5 days; severe cerebral injury (intraventricular hemorrhage ≥ stage III according to Papile’s definition or periventricular leukomalacia diagnosed on head ultrasound or cerebral magnetic resonance imagery); bronchopulmonary dysplasia, defined as the need for oxygen or respiratory support at day 28; catheter-related bloodstream infections, defined as at least one positive blood culture associated with clinical manifestations or antibiotic use for at least 5 days concomitant with a central line; and length of hospital stay.

### 2.5. Statistical Analysis

Quantitative and qualitative variables were compared using chi-square or Fisher’s exact tests, and Student’s t-test or Kruskal–Wallis tests (when conditions for Student’s *t*-test were not satisfied), respectively. The association between the use of 3-in-1 STD-PN and RBWD DoL1-7 was evaluated using a generalized logistic regression analysis (PROC GLIMMIX in SAS^®^ v9.2, SAS Institute, Cary, NC, USA), with the following confounding factors: GA, SGA, sex, RDS, early neonatal infection, use of central venous line, PN duration, and enteral intakes on DoL3. This analysis included the neonatal unit as a random effect to consider the correlation of data owing to the center effect. The results are presented as the beta (standard deviation [SD]). Statistical significance was set at *p* < 0.05.

## 3. Results

During the study period, of the 835 infants born with a GA between 32^0/7^ and 34^6/7^ SA and admitted to the 21 neonatal units of PACA-Corse-Monaco, 100 were excluded because of death (n = 6), congenital anomalies (n = 2), and relevant missing data (n = 92). Among the remaining 735 infants, 315 (42.8%) infants received PN from birth up to DoL3 and were included in the analyses. According to gestational age, 70.5% of 32-week-old infants (120 on 170), 54% of 33-week-old infants (108 on 200), and 23.8% of 34-week-old infants (87 on 365) received PN at DoL3 (*p* < 0.0001). Among them, 178 (56.5%) received 3-in-1 standardized PN (3-in-1 STD-PN).

Table 1 summarizes the perinatal characteristics of the study population. Gestational age and early neonatal morbidities, including RDS and early neonatal sepsis, did not significantly differ between the two groups of infants. Infants managed with 3-in-1 STD-PN were more likely to be SGA at birth (41 (23%) vs. 18 (13.1%); *p* < 0.01).

In the Other-NP study group, 84 infants (61.3%) received a 2-in-1 solution alone, 25 infants (18.2%) received a 2-in-1 solution with Y-site addition of glucose, 1 infant (0.7%) received a 2-in-1 solution with Y-site amino acid, 5 infants (3.6%) received a 2-in-1 solution with Y-site lipid emulsion and vitamins, and finally, 22 infants (16%) received simple PN solutions (glucose with electrolytes).

### 3.1. Enteral and PN Intakes

Table 2 presents the comparison of parenteral and total nutritional intake between the two groups of infants during the first week of life. Breastfeeding was the principal enteral nutrient of the population and was not significantly different between the two groups (58.4% vs. 56.2% in infants who received or did not receive 3-in-1 STD-PN at DoL3, *p* = 0.38). The mean parenteral AA and energy intakes were consistently higher at DoL1, 3, and 7 in MP infants who received 3-in-1 STD-PN than in infants from the Other-PN solution group. These differences were not compensated for by enteral intake.

On DoL3, in contrast to their counterparts, all infants with 3-in-1 STD-PN received protein (100% vs. 83%, *p* = 0.99) and lipids (100% vs. 3.6%, *p* < 0.0001). The mean parenteral AA and energy intakes were, respectively 85% (*p* < 0.0001) and 27% (*p* < 0.0001) higher in infants receiving 3-in-1 STD-PN. Solutions of 3-in-1 STD-PN provided higher energy content with energy intake/volume of PN solutions ratio (kcal/mL) significantly higher than the other PN solutions (0.54 ± 0.1 vs. 0.45 ± 0.05 kcal/mL at DoL3, *p* < 0.0001), whereas use of central line was less frequent (32% vs. 37.9%, *p* < 0.05).

On DoL7, 86 infants needed PN and 57 received 3-in-1 STD-PN. Infants who received higher parenteral energy and AA intakes were managed with 3-in-1 STD-PN. Energy and AA were 43% and two-fold higher in this infant group (Table 2).

### 3.2. Type of PN and Early Neonatal Weight Growth

Infants receiving 3-in-1 STD-PN lost 36% less BW during the first week of life, with median RBWD DoL1-7 of −2.5% vs. −3.9% in Other-PN infants (*p* < 0.05). The proportion of infants with RBWD ≥ 90th percentile was higher in those who received 3-in-1 STD than in their counterparts [27 (15.2%) vs. 6 (4.4%) infants, *p* < 0.01] (Table 2).

Table 3 presents the multivariable analysis results, wherein a positive beta value indicates a lower relative BW difference during the first week of life. Adjusted to confounding factors, including GA, nCPAP use, infection, and nutritional intakes at DoL3, RBWD DoL1-7 was significantly less in the 3-in-1 STD-NP group than their counterparts with beta (SD) = 2.08 (0.91) (*p* = 0.02).

## 4. Discussion

In this study, which included MP infants with GA between 32^0/7^ and 34^6/7^ weeks who required PN, use of ready-to-use 3-in-1 standardized PN solutions from birth onwards demonstrated higher AA and caloric intake and improved early neonatal growth compared to that observed in the control group.

The ready-to-use standardized PN formulations used in our study were conceptualized after a close discussion between neonatologists and pharmacists regarding the common nutritional needs of preterm infants. In the study population, clinicians could use three different STD-PN solutions from birth onwards, according to postnatal growth phases, clinical conditions, nutritional needs, and the type of venous line access (central or peripheral). These were 3-in-1 solutions containing electrolytes, micronutrients, and vitamins without additional complements. Compared with the simple or 2-in-1 solutions (AA and glucose) used in our study, 3-in-1 STD-PN solutions provided systematically higher caloric and protein intake from birth onwards and avoided nutritional deficiency, particularly in macronutrients. For instance, 100% of infants in the 3-in-1 STD-PN group received lipids and AA from birth onwards, whereas approximately 90% of their counterparts did not receive intravenous lipids and approximately 15% did not receive intravenous AA on DoL3. Our findings confirm the benefits of using standardized PN solutions that optimize nutritional intake and limit prescription errors and oversights [14]. The use of these PN solutions would have the potential to standardize nutritional practice and improve both the quality and process of nutritional care and patient outcomes in the region. This strategy has been adopted by several neonatal societies [15,16,17] to harmonize nutritional intake and avoid nutritional deficiencies. This study provides further information to help determine PN formulations suitable for premature infants; however, the optimal formulation remains debatable.

Compared to term newborns, MP infants are a vulnerable population requiring higher energy and protein intake to ensure optimal growth of various organs, including the brain. They constitute an overlooked population with a high risk of neonatal growth failure [8,9,18,19]. Growth failure, with an incidence of approximately 30%, occurs soon after birth and is linked to nutritional intake insufficiency from birth onwards. We previously identified the association of neonatal growth failure with low energy and protein intake during the first week of birth [9]. However, the best method of feeding MP infants has not been identified in the literature. Hence, nutritional management and feeding practices widely vary between neonatal centers [8,11,18,20]. Nutrition practice and compliance with guidelines of MP infants vary amongst neonatal clinicians, perhaps due to the lack of evidence from randomized trials on which to base clinical practice [20]. Some MP infants require PN to avert or treat hypoglycemia or to provide sufficient nutritional intake when enteral feeding is delayed or impossible. In the literature, the rate of PN requirements widely varies from 5% to 90% depending on GA and clinical practices, and the duration is generally short [8,20,21,22]. Moreover, PN generally provides AA and lipid intake at the lower limits of current recommendations, partly because of the use of peripheral venous lines with limited concentrated solutions. In our study, PN was mostly administered via peripheral venous line (65%) for a short duration. Even over a short period, PN may provide additional nutrients to prevent excessive weight loss [23,24]. The first week’s prescribed protein intake is a key determinant of weight gain and head growth from birth to term age in MP infants [24]. We previously demonstrated that energy and protein deficits during the first week of life are associated with neonatal growth failure [9]. Indeed, in this last study, we observed a 46% reduction in extrauterine growth restriction for each 1 g/kg/d increase in protein intake at the end of first week of life. Moreover, for each 10 kcal/kg/d increase in energy at the end of the first week of life, we observed 27% lower odds of extrauterine growth restriction at discharge. Minimal postnatal weight loss in healthy moderately premature infants has been shown to be associated with earlier enhancement of nutrient intake [18]. Also, the use of early PN reduces early weight loss and the time taken to return to birth weight in preterm infants [1]. In a randomized controlled trial of 92 infants born with GA between 30^0/7^ and 33^6/7^ weeks with a peripheral venous access line, infants receiving adapted ternary PN had higher protein and caloric intake and higher growth velocity from birth to day 21 than those managed with 10% glucose solution [23]. Providing only carbohydrates in the first few days following birth will probably lead to protein deficiency. In our study, we also observed limited early BW loss in infants managed with a strategy that provided high macronutrient intake from birth onwards [9]. This association did not result from the type of venous access, since the use of the peripheral venous line was more frequent in infants managed with 3-in-1 STD PN. High-osmolality solutions are more likely to induce phlebitis [25]. In our study, the osmolarity of the 3-in-1 STD-PN solutions 1 and 2 is less than 900 mOsm (Appendix A); so peripheral venous delivery should be used. A central venous line is generally required to maintain long-term venous access. Avoiding the use of a central venous line allows us to reduce infection and other associated complications [25]. Altogether, these findings provide evidence of the importance of optimizing early nutritional intake during the transition period in MP infants who require PN.

This study has some limitations, especially in relation to the protocol design. We did not collect data on daily nutritional intake and BW parameters; we could not evaluate the impact of the PN strategy on cumulative protein and energy intake during the first week of life and on maximal BW loss. We were also unable to evaluate the effects of sodium intake on growth. Finally, groups of infants who did not receive 3-in-1 STD-PN were managed using different PN solutions available in the neonatal intensive care units in our region. Further studies are required to compare this strategy to other three-in-one standardized PN solutions, which have been demonstrated to improve nutritional intake in preterm infants compared to individualized strategies; however, specific manipulation with the addition of extra fluids, micronutrients, and vitamins is required [26,27]. The main strength of this observational population-based study is that it demonstrated a wide variation in nutritional practices in real-world conditions in a large cohort of MP infants.

## 5. Conclusions

In this study of MP infants who received PN from birth onwards, we confirmed that use of three-in-one standardized parenteral solutions provided higher energy and AA intake and limited early excessive weight loss. This strategy contributes to optimizing the nutritional intake and limiting errors. Our findings highlight the importance of better nutritional intake in infants who require PN, regardless of whether PN is administered for a short period of time or through peripheral venous access. This study provides further information to help determine PN formulations suitable for premature infants; however, the optimal formulation remains debatable.

## Figures and Tables

**Table 1 nutrients-16-01292-t001:** Characteristics of the study population.

Characteristics, n (%)	Overall Population n = 315	3-in-1 STD-PN n = 178	Other-PN n = 137	*p*
**Obstetrical characteristics**				
Cesarean delivery	193 (61.2)	113 (63.4)	80 (58.3)	0.78
Prenatal steroids	263 (83.4)	150 (84.2)	113 (82.4)	0.23
Multiple pregnancy	114 (36.1)	66 (37.0)	48 (35.0)	0.85
Pre-eclampsia	97 (30.7)	64 (35.9)	33 (24.0)	0.02
PROM	37 (11.2)	27 (15.1)	10 (7.3)	0.03
Chorioamnionitis	6 (1.9)	1 (0.5)	5 (3.6)	0.09
**Neonatal characteristics**				
GA (weeks) *	32.8 ± 0.8	32.9 ± 0.7	32.8 ± 0.8	0.17
Sex male	164 (52.1)	94 (52.8)	70 (51.0)	0.36
Birth weight (g) *	1841 ± 353	1817 ± 372	1871 ± 325	0.02
SGA	59 (18.7)	41 (23)	18 (13.1)	0.004
RDS	39 (12.4)	22 (12.3)	17 (12.4)	0.34
nCPAP	224 (71.1)	131 (73.6)	93 (67.8)	0.71
Early onset sepsis	15 (4.7)	9 (5.1)	6 (4.3)	0.99
NEC, stage ≥ II	11 (3.4)	8 (4.4)	3 (2.2)	0.71
Severe brain injuries	4 (1.2)	1 (0.5)	3 (2.1)	0.54
Central venous line	109 (34.6)	57 (32)	52 (37.9)	0.044
CRBI	6 (1.9)	5 (2.8)	1 (0.7)	0.98
LOS (days) *	27.3 ± 11.0	27.8 ± 12.1	26.7 ± 9.5	0.009
NICU admission	91 (28.8)	58 (32.5)	33 (24.0)	0.002
NP duration (days) ^†^	6 (4, 8)	6 (5, 8)	5 (4, 7)	0.02

Legend: 3-in-1 STD-PN, 3-in-1 standardized parenteral nutrition; CRBI, catheter-related bloodstream infections; GA, gestational age; LOS, length of stay; nCPAP, nasal continuous positive airway pressure; NEC, necrotizing enterocolitis; NICU, neonatal intensive care unit; PROM, premature rupture of membranes; RDS, respiratory distress syndrome; SGA, small for gestational age. * Mean ± standard deviation; ^†^ Median (IQR).

**Table 2 nutrients-16-01292-t002:** Parenteral and total nutritional intakes during the first week after birth.

Nutritional Intakes (Mean ± SD)	Overall Population n = 315	3-in-1 STD-PN n = 178	Other-PN n = 137	*p*
**Early neonatal growth**				
RBWD DoL1-7 (%) *	−3 (−3.7, 0.3)	−2.5 (−5.9, 1.1)	−3.9 (−8.5, −0.5)	0.02
RBWD DoL1-7 ≥ 90 thp n (%) ^†^	33 (10.5)	27 (15.2)	6 (4.4)	0.002
**Nutritional intakes DoL1**				
Energy parenteral (kcal/kg/d)	34 ± 7	36 ± 6	30 ± 6	<0.0001
AA parenteral (g/kg/d)	1.2 ± 0.6	1.6 ± 0.4	0.7 ± 0.5	<0.0001
Energy Tot (kcal/kg/d)	42 ± 11	46 ± 11	37 ± 9	<0.0001
Protein Tot (g/kg/d)	1.5 ± 0.7	1.8 ± 0.6	0.9 ± 0.5	<0.0001
**Nutritional intake DoL3**				
Energy parenteral (kcal/kg/d)	34 ± 13	37 ± 13	29 ± 11	<0.0001
AA parenteral (g/kg/d)	1.3 ± 0.7	1.5 ± 0.6	0.8 ± 0.5	<0.0001
Energy Tot (kcal/kg/d)	66 ± 15	69 ± 12	62 ± 18	0.009
Protein Tot (g/kg/d)	2.2 ± 0.7	2.4 ± 0.6	1.8 ± 0.8	<0.0001
**Nutritional intake DoL7**				
Energy parenteral (kcal/kg/d) ^‡^	48 ± 24	53 ± 25	37 ± 13	<0.001
AA parenteral (g/kg/d) ^‡^	1.7 ± 0.9	2 ± 0.9	1 ± 0.7	<0.0001
Energy Tot (kcal/kg/d)	90 ± 20	93 ± 20	87 ± 18	0.01
Protein Tot (g/kg/d)	2.7 ± 1	2.8 ± 1	2.5 ± 1	0.05

Legend: AA parenteral: parenteral amino-acids intakes; energy parenteral: parenteral energy intake; Energy Tot: total energy intakes (parenteral and enteral); Protein tot: total protein intakes (parenteral and enteral); RBWD DoL1-7: relative bodyweight difference between day of life 1 and 7; RBWD DoL1-7 ≥ 90th p: proportion of infants with a relative bodyweight difference between day of life 1 and 7 above the 90th percentile (calculated at +5% of the overall population); 3-in-1 STD-PN: 3-in-1 standardized parenteral nutrition; * Median (IQR). ^†^ Number (%). ^‡^ The number of infants who received PN at DoL7 was as follows: overall population (n = 86), infants with 3-in-1 STD-PN (n = 57), and infants with other parenteral nutrition received (Other-PN) (n = 29).

**Table 3 nutrients-16-01292-t003:** Factors associated with relative body-weight difference between day of life 1 and 7 (RBWD DoL1-7).

	Univariate			Multivariate		
Variables	Parameter β	SD	*p*	Parameter β	SD	*p*
3-in-1 STD-PN	2.30	1.02	0.02	2.08	0.91	0.02
GA						
32 weeks	−1.38	0.83	0.09	−0.41	0.83	0.62
33 weeks	0.89	0.83	0.28	0.97	0.79	0.22
34 weeks	Reference			Reference		
Female sex	−0.41	0.67	0.54	0.04	0.63	0.95
SGA	4.59	0.79	<0.0001	3.26	0.91	<0.001
Central line	2.63	0.71	<0.001	2.64	0.85	0.002
NICU	0.22	1.39	0.87	−1.24	1.26	0.33
PN duration	0.27	0.25	0.28	−0.18	0.31	0.55
Enteral EI (DoL3)	0.05	0.02	0.01	0.07	0.02	0.001
nCPAP	−2.97	0.71	<0.0001	−2.37	0.70	<0.001
Early-onset infection	0.98	1.57	0.53	2.80	1.47	0.06

Legend: 3-in-1 STD-PN, 3-in-1 standardized parenteral nutrition; Enteral EI (DoL3), enteral energy intake (kcal/kg/day) on day 3; GA, gestational age; nCPAP, nasal continuous positive airway pressure; NICU: neonatal intensive care unit; SGA, small for gestational age.

## Data Availability

The original contributions presented in the study are included in the article/Appendix A, further inquiries can be directed to the corresponding author. The Regional Perinatal Network Database can only be accessed through a secure server after obtaining authorization from the Scientific Committee of the Regional Perinatal Health Network. The data are available upon request from the authors.

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
