# Peer review of "Association of Standardized Parenteral Nutrition with Early Neonatal Growth of Moderately Preterm Infants: A Population-Based Cohort Study"

_nutrients, 2024, doi:10.3390/nu16091292_

Round 1
Reviewer 1 Report
Comments and Suggestions for Authors
Thank you for submitting the manuscript "Association of standardized parenteral nutrition with early neonatal growth of moderately preterm infants: a population-based cohort study" to Nutrients. The manuscript contains relevant information regarding the use of PN in infants. However, I believe that one of the information that readers can seek is related to the amount of PN used for a given age of infant for how long to improve outcomes. Additionally, I have other considerations:
Line#33: what does this number under writing mean?
Line#51: why are lipids usually administered separately? An explanation needs to be added.
Line#73: Was the study non-interventional? I believe that the study was carried out blindly, but for the patient there was intervention.
Line#82: because the infants who died were excluded. Malnutrition is also an aggravating factor that can contribute to the depression of the infant's general condition and lead to death.
Line#108: what is GA? Add a definition.
- It is important to add to the methodology what was used in the infants from the other group. If it was the Nutrients separated if it was a 2-1. State the percentage of each case within this group.
Line#207: compared to the control group?
Line#208: this has already been described in item M&M
- The discussion of the manuscript is initiated based on the fact that the results of PN 3-in-1 are better than the others. But isn't that obvious? Won't the infant who has access to more nutrients have a better outcome? Or did the infants who received the other PN receive another type of supplementation?
Line#252: I believe that the most important thing about this claim is also adding amounts of PN. It would be interesting to add a list of data from the manuscript with weight gain, survival rate, amount ingested in the first week, etc.
Reviewer 2 Report
Comments and Suggestions for Authors
I believe this is an interesting manuscript on a well-conducted study on parenteral nutrition in preterm infants. Even though the study design is cohort and not RCT, I believe these results add value to current knowledge on this topic. The tables are nicely presented and references are up to date. I believe this manuscript will be of interest to a broad audience.
I only have one small comment, to improve your manuscript I would suggest adding the study setting and participant description (number eg) to the abstract.
Author Response
Thank you for your reply.
We have added elements of study method (line 32) and results (line 34) into the abstract.